# Dissecting the Structural Dynamics of Authentic Cholesteryl Ester Transfer Protein for the Discovery of Potential Lead Compounds: A Theoretical Study

**DOI:** 10.3390/ijms241512252

**Published:** 2023-07-31

**Authors:** Yizhen Zhao, Dongxiao Hao, Yifan Zhao, Shengli Zhang, Lei Zhang, Zhiwei Yang

**Affiliations:** MOE Key Laboratory for Nonequilibrium Synthesis and Modulation of Condensed Matter, School of Physics, Xi’an Jiaotong University, Xi’an 710049, China; zyz9856@stu.xjtu.edu.cn (Y.Z.); dongxiaohao18@163.com (D.H.); zyfan0911@outlook.com (Y.Z.); zhangsl@xjtu.edu.cn (S.Z.); zhangleio@xjtu.edu.cn (L.Z.)

**Keywords:** cholesteryl ester transfer protein (CETP), authentic CETP (CETP^Authentic^), molecular dynamics simulation, CETP conformational dynamics, CETP inhibitor

## Abstract

Current structural and functional investigations of cholesteryl ester transfer protein (CETP) inhibitor design are nearly entirely based on a fully active mutation (CETP^Mutant^) constructed for protein crystallization, limiting the study of the dynamic structural features of authentic CETP involved in lipid transport under physiological conditions. In this study, we conducted comprehensive molecular dynamics (MD) simulations of both authentic CETP (CETP^Authentic^) and CETP^Mutant^. Considering the structural differences between the N- and C-terminal domains of CETP^Authentic^ and CETP^Mutant^, and their crucial roles in lipid transfer, we identified the two domains as binding pockets of the ligands for virtual screening to discover potential lead compounds targeting CETP. Our results revealed that CETP^Authentic^ displays greater flexibility and pronounced curvature compared to CETP^Mutant^. Employing virtual screening and MD simulation strategies, we found that ZINC000006242926 has a higher binding affinity for the N- and C-termini, leading to reduced N- and C-opening sizes, disruption of the continuous tunnel, and increased curvature of CETP. In conclusion, CETP^Authentic^ facilitates the formation of a continuous tunnel in the “neck” region, while CETP^Mutant^ does not exhibit such characteristics. The ligand ZINC000006242926 screened for binding to the N- and C-termini induces structural changes in the CETP unfavorable to lipid transport. This study sheds new light on the relationship between the structural and functional mechanisms of CETP. Furthermore, it provides novel ideas for the precise regulation of CETP functions.

## 1. Introduction

Cholesteryl ester transfer protein (CETP) encoded by the CETP gene, is a 476-residue-long plasma glycoprotein that plays a crucial role in facilitating the hetero exchanges of cholesteryl esters (CE) and triglycerides (TG) [1]. The development of hyperalphalipoproteinemia 1, a monogenic condition with autosomal dominant inheritance, has been linked to pathogenic variants in the CETP gene, highlighting the crucial function of CETP in high-density lipoprotein metabolism [2,3,4]. CETP redirects CE from high-density lipoprotein (HDL) to more atherogenic low-density lipoprotein (LDL) and very-low-density lipoprotein (VLDL) [5]. Blocking CETP to inhibit proatherogenic lipoprotein remodeling represents a promising strategy for mitigating cardiovascular diseases (CVD), particularly atherosclerosis cardiovascular disease (ASCVD), which is the leading cause of death in many countries [6]. Therefore, a large number of studies have been devoted to the pharmacologic inhibition of CETP, and abundant preclinical evidence strongly indicates the anti-atherogenic behaviors associated with CETP deficiency and inhibition [6,7]. However, it should be noted that some studies have identified CETP agonists capable of improving cardiovascular disease [8]. As a result, the role of CETP in cardiovascular health, whether it exhibits anti-cardiovascular or cardiogenic effects, may require evaluation on an individual basis, considering the specific metabolic system involved.

In light of the current intense clinical interest in the functional regulation of CETP, there is a pressing need to gain a deeper understanding of its structure and function, particularly regarding the molecular basis for the lipid transfer mechanism [6,9,10]. X-ray crystallographic analyses showed that CETP resembles a ‘banana’ shape with N- and C-terminal β-barrel domains, a central β-sheet, a C-terminal extension (Helix X, Glu465-Ser476), and a ~60 Å long unconnected hydrophobic central cavity [11]. Subsequent electron microscopy (EM) images combined with molecular dynamics (MD) simulations showed the flexibility of the distal part of the β-barrel structure [12], wherein the N-terminal β-barrel domain displays considerably higher flexibility in solution compared to mutant CETP and the distal region of the C-terminal β-barrel domain undergoes extension. In addition, the research exhibited that CETP’s N-terminus penetrates the HDL surface and core (~50 Å), whereas, for the C-terminus, only the surfaces of LDL and VLDL (~25/20 Å) [12,13,14,15] are penetrated. During the formation of the ternary complex, 10° of tilting within the β-barrel strands causes the connection of the CETP central cavity, thereby forming a hydrophobic tunnel for neutral lipid exchange between different lipoprotein fractions [13,14]. Aside from the tunnel hypothesis described above, there are two other theories: (a) shuttle mechanism, where CETP carries neutral lipids by transiently associating with HDL or LDL through the regulatory role of Helix X [9,16]; (b) a modified tunnel mechanism involving the formation of a CETP dimer [17]. At present, there is still no single conclusion about the mechanism of CETP-mediated lipid exchange [5,15].

Experimental studies have demonstrated that inhibition of CETP reduces cholesterol uptake by cells within atherosclerotic plaques and enhances cholesterol effluvia [18]. CETP-deficient patients also exhibit increased HDL concentrations [19], and animal studies have shown that CETP inhibition leads to higher HDL levels and reduced ASCVD [20,21]. Several inhibitors, including torcetrapib, dalcetrapib, evacetrapib, anacetrapib, and obicetrapib, have been developed, which have shown the potential to increase plasma HDL-C levels and/or decrease LDL-C levels, thereby reducing ASCVD events [20]. To better understand the inhibition mechanism, molecular dynamics simulations have been employed to study the interactions of inhibitors with CETP. Previous studies have revealed that anacetrapib may destabilize the CETP–lipoprotein complex, leading to a reduction in CE transport [22]. Chirasani et al. proposed that the inhibitors torcetrapib and anacetrapib inhibit CETP by binding to the CETP tunnel center through hydrophobic interaction forming a physical occlusion of the hydrophobic channel [23]. Our previous studies have indicated that the inhibitors torcetrapib, anacetrapib, and evacetrapib reduce the flexibility of N- and C-termini and Helix X, along with the stability of hydrophobic channels, and also inhibit the formation of the CETP N-terminal opening [24]. However, clinical trial evaluations have shown that some inhibitors cause unacceptable side effects such as increased systolic blood pressure, increased aldosterone and cortisol synthesis, and non-target toxic effects, while others are ineffective in treating ASCVD [20]. Therefore, the structural characteristics of CETP and lipid transfer mechanisms need to be further investigated in order to develop more effective CETP inhibitors for ASCVD treatment.

As noted above, recent X-ray, EM, and MD studies have revealed several important physical attributes of CETP and a substantial molecular basis for neutral lipid transport mediated by CETP [11,12,14]. However, almost all of the evidence stems from a fully active mutation (mutant c444 construct, C1A C131A N88D N240D N341D), which is constructed to yield superior crystals [11,25]. Approximately 30 CETP variations have been recognized as clinically harmful among the numerous CETP variants (https://www.ncbi.nlm.nih.gov/clinvar/, accessed on 25 July 2023) [26]. Moreover, studies on insertion and missense mutagenesis have revealed that mutations with disordered structures (e.g., the mutations on amino acids 48–53, 373–379, and 470–475) have profound effects on lipoprotein selectivity and CE transferability [27,28,29]. Additionally, the first four residues (C1 S2 K3 G4) are absent in the available crystal structures [11,30]. The research relying on the mutant structure may not fully capture the dynamic physiological processes involved in lipid transport mediated by CETP under normal physiological conditions. Thus, the determination of the authentic CETP structure is an urgent problem for researchers, especially to elucidate whether the structural differences between authentic CETP and the mutant c444 construct influence the functional effect of CETP. Moreover, gaining insights into the structural and dynamic characteristics of CETP in physiological solutions could greatly contribute to the development of novel CETP inhibitors. In our previous studies, an integrative approach was adopted to dissect the association between CETP and its ligands, which provides notable insights into the dynamic behaviors of CETP [13,24,31]. In this work, the aforementioned computational tools were utilized to evaluate the spatial distinctions between CETP^Authentic^ and CETP^Mutant^, with a primary focus on stability, flexibility, hydrophobicity, and residue reorientation, all of which are closely related to the lipid transfer function. In addition, virtual screening and MD simulations were performed based on the authentic CETP structure against the ZINC library [32]. The most promising ligands were selected by MM/GBSA energy calculation and structural analysis, and the dynamic changes in CETP structure induced by ligand binding were analyzed. We anticipate that the results may aid in the understanding of the CETP-mediated lipid transfer mechanisms.

## 2. Results and Discussion

### 2.1. Contrast in Structural Stability and Flexibility

The equilibration reliability of each system was first verified by monitoring the time evolution of structural parameters [33]. The backbone-atom root-mean-square deviation (RMSD) curves of CETP^Mutant^ and CETP^Authentic^ reach a plateau after approximately 40 ns, oscillating with minor variations (Appendix A), suggesting that the simulations are sufficient to obtain the stability of each system. Note that all the analyses were performed over three replicate simulations to ensure the consistency of each individual simulation. The motion differed from previous MD simulations of CETP^Mutant^ complexed with anacetrapib, where the backbone-atom RMSD rapidly reach equilibrium within a 20 ns MD simulation, probably due to the inhibitor-mediated CETP configuration alterations [34]. The time-evolution radius of gyration (Rg) further confirms the equilibrium state, with the values of CETP^Mutant^ and CETP^Authentic^ converged to 34.2 ± 0.1 Å and 33.5 ± 0.1 Å (Appendix A).

As far as we are aware, structural flexibility is essential and beneficial for the CETP configuration adjustment which is associated with lipid transfer [13,35]. There are seven important flexible regions of the CETP structure: residues Asp290–Gln318 (flap Ω1), Pro351–Ser358 (flap Ω2), and Lys392–Ser404 (flap Ω3) of the C-terminal end, and, with respect to the N-terminal barrel domain end, Glu46–Val55 (flap Ω4), Gly100–Gln111 (flap Ω5), and Phe155–Trp162 (flap Ω6). In addition, Helix X (residues Glu465–Ser476) is also relatively flexible (Figure 1) [13]. In this work, the C_α_ atom RMSF values of the seven regions were calculated as an indicator of structural flexibility [13,36]. As shown in Appendix A, the seven regions of CETP^Mutant^ and CETP^Authentic^ are indeed flexible, consistent with previous atomistic MD simulations [9,13,15], while the peak RMSF values of flap Ω3 and Helix X in CETP^Authentic^ are higher than those of CETP^Mutant^, especially for flap Ω3 (Appendix A). In contrast to CETP^Mutant^, CETP^Authentic^ is much more flexible in the regions of flap Ω3 and Helix X, and its C-terminal end might have a distinct configuration adjustment. Taken together, the equilibrium of these simulations is reliable, and CETP^Authentic^ is flexible, in accord with the experimental fact that it is difficult to crystallize the native CETP, and importing some specific mutations can improve the crystal quality [11].

The distinct configuration adjustments were further explored by the superimposition of typical conformations from MD simulations. The initiating conformations of CETP^Mutant^ and CETP^Authentic^ perfectly overlap each other, except for residues Ala1–Gly4 (Figure 1). With respect to equilibrium conformations, they experience configuration changes through different processes. Compared to the initiating conformations, the N-terminal domain of CETP^Mutant^ stretches longer along the axis and deviates from Helix X, while the C-terminal domain of CETP^Authentic^ partly flexes back to Helix X. Indeed, the equilibrium conformation of CETP^Mutant^ stretches itself by 12.4°, while the authentic one huddles itself up by 4.5° (Appendix A). In contrast to the equilibrium structure of CETP^Mutant^, the authentic N-terminal and C-terminal domains take an oblique route to Helix X, with bending angles increased by 25.1° and 12.7°, respectively (Appendix A). Regarding the thermodynamic motion of neutral lipids and the CETP hydrophobic tunnel, the relatively straightening curvature of the CETP concave face might be beneficial for CETP binding and lipid exchange, as supported by electron micrographs and MD simulations of CETP–liposome complexes [14,15]. Hence, the huddled conformation of CETP^Authentic^ may be disadvantageous for liposome binding and potentially interfere with lipid transfer. The decline of Rg curves of CETP^Authentic^ could also suggest an unfavorable factor. Although the equilibrium conformation of CETP^Authentic^ is not as flat as that of CETP^Mutant^, its average expansion rate of internal cavity volume is relatively greater, with values of 6.4% and 1.5%, respectively. In previous MD simulations, we found that internal hydrophobic cavities of CETP^Mutant^ (mutant c444 construct) are generally stable in an aqueous solution, which may serve as indirect evidence supporting the tunnel hypothesis [12,13]. In this study, the rather obvious enlargement of the internal cavity volume of CETP^Authentic^ further suggests the possibility of tunnel transfer.

### 2.2. Contrast in the N-Terminal End, C-Terminal End, and “Neck” of the Hydrophobic Tunnel

Atomistic MD simulations and EM data confirm that the N-terminal barrel domain end of CETP^Mutant^ can penetrate the HDL particle surface through the major anchoring role of residues Trp105, 106, and 162, associated with the large-scale structural alterations in the N-terminal end (flaps Ω4–6) that give rise to an opening (~11 Å, N-opening) for the hetero exchanges of neutral lipids [14,15]. Upon our explicit solvent MD simulations, distance analysis between residues Trp 106 and Trp162 further demonstrates the formation of an N-opening (Figure 2). In the initiating conformations of CETP^Mutant^ and CETP^Authentic^ (0 ns), the maximum distances between Trp106 and Trp162 (d_Trp106-Trp162_) are 16.30 Å and 16.35 Å, respectively. After 30 ns, they reduce slightly, while subsequently reaching plateaus (15.57 Å and 15.92 Å) at 80 ns (Figure 2). The values of d_Trp106-Trp162_ remain constant at ~16 Å for both CETP^Mutant^ and CETP^Authentic^ during 100 ns MD simulations, consistent with our previous MD results showing that the CETP^Mutant^ N-terminal end is intrinsically flexible and its overall surface hydrophobicity is stable in solution [13]. Meanwhile, the larger size of the N-opening (~16 Å) in our simulations (in aqueous solutions) is reasonable to consider in that the N-terminal end of CETP could roughly be regarded as a hydrophobic region, and the hydrophobic cholesteryl oleate molecules of HDL will relatively enhance its stability by promoting the formation of intramolecular hydrogen bonds [13,14].

Differing from the N-terminal end, the C-terminal distal region (flaps Ω1–3) of CETP^Mutant^ has a more globular configuration, with less hydrophobicity and an unstable hydrophobic surface pore P1 [13]. In P1 there is observed shrinkage during previous 8.5 ns MD simulations, with the formation of a substantially hydrophobic pore [13]. The motion might indicate an opening (C-opening) for lipid traverse, with interior residues Phe301, Met412, and Ile413 playing a key role in this process [13,37,38]. Compared to the distal portion of the N-terminal domain, the C-terminal end exhibits larger C_α_ fluctuations, with an increase in overall RMSF (Appendix A) and a relatively large exchange of hydrophilic/hydrophobic SASA, for both CETP^Mutant^ and CETP^Authentic^. During the MD simulations, the total SASA of the C-terminal end in CETP^Mutant^ increases by about 4%, while the authentic one has a larger increasing rate of ~8%. Consistent with previous MD results [13], the hydrophobic SASA of the C-terminal end in CETP^Mutant^ decreases (~2%); however, the opposite occurs for the authentic one (~8% ↑) (Table 1). The data suggest that hydrophobic SASA occupies a dominant role in the considerable surface changes of the two C-termini (“Mutant” and “Authentic”), and their configurations should be different in aqueous solutions because of the generally more exposed surface of CETP^Authentic^. The maximum distance between residues Phe301 and Met412 (d_Phe301-Met412_) has been used to further estimate the structural differences. As Figure 3 shows, d_Phe301-Met412_ of CETP^Authentic^ is a constant ~10 Å during 100 ns MD simulations, with residue Phe301 slightly deviating from residue Met412. By contrast, the mutant one reaches a plateau at ~25 ns with a decrement of ~2 Å. In the beginning, residue Phe301 in CETP^Mutant^ orients towards the backbone of residue Met412. Then, it gradually gets close to residue Met412 in the first ~25 ns, with d_Phe301-Met412_ leveling off at 7.9 Å. This is in accord with the aspirations of hydrophobic SASA in that the “C-opening” region of CETP^Authentic^ retains hydrophobicity and is more conducive to the development of an opening. The spatial size of CE’s rigid steroid ring (about 4 × 6 Å, TG is much larger) and the van der Waals radius of the hydrogen atom (~2.4 Å) [11] suggest that they require at least 10 Å to induce a potential opening (“C-opening”), enabling CE to smoothly enter LDL or VLDL without orientation adjustment [5,12]. Hence, an additional orientation adjustment of lipids is required for the exchange through the C-terminal domain of CETP^Mutant^, leading to an obstruction of CETP-mediated lipid exchange. However, this situation does not occur in the authentic case, because of its good hydrophobicity and larger value of d_Phe301-Met412_. Furthermore, residues Phe301 and Met412 are located in flexible flapΩ1 and stable α-helix, respectively. During the simulations, the hydrophobic area of flap Ω1 increases by 4% in CETP^Authentic^, while it decreases by 13% in CETP^Mutant^ (Table 1). This suggests that flapΩ1 of CETP^Authentic^ tends to extend in physiological solution, while flapΩ1 of CETP^Mutant^ prefers to embed within the protein, agreeing well with previous conclusions that the authentic C-terminal end is more exposed to the solvent and has a larger value of d_Phe301-Met412_, further supporting the possibility of a C-opening. The interesting thing is that mutations of residues 54 and 106 at the N-terminus, and residue 318 at the C-terminus are associated with hyperalphalipoproteinemia 1, where Trp106 is an important residue for CETP–HDL interaction, and therefore these mutations may alter the structure and hence the function of CETP (https://www.ncbi.nlm.nih.gov/clinvar/, accessed on 25 July 2023) [15,26].

The amphipathic Helix X of CETP is located about 50~60 Å from the N-terminal domain end, and its closed state permits a more sustained CE transfer [14,15,39]. Compared with the mutant case, Helix X in CETP^Authentic^ is more flexible and exposed to solvent, with a larger increase in hydrophobic SASA (~9%) (Appendix A and Table 1). However, both of them have an approximate configuration and steadily maintain the closed state during the 100 ns MD simulations (Appendix A). This proposal agrees well with EM structural studies and MD simulations showing that the closed state of Helix X should be a necessary condition for proper lipid transfer functioning [13,14,15]. Our previous MD simulations revealed that separated cavities in CETP^Mutant^ become connected to each other and to the central cavity, forming a continuous hydrophobic tunnel for CETP-mediated lipid transfer [13]. And there exists a potential barrier, consisting of residues Phe265 and Met433, in the “neck” region of CETP [13,14,38]. Any mutation that reduces the size of the “neck”, is expected to impair the transfer activity of CETP [11]. For example, CETP with mutation Ile443Trp observably lacks transfer competence [11]. Upon a potential barrier, examination of the maximum distance between residues Phe265 and Met433 (d_Phe265-Met433_) allows an estimation of the size of the “neck”. As Figure 4 shows, the d_Phe265-Met433_ is constant around 16.5 Å in CETP^Authentic^, while it decreases to ~10.5 Å in CETP^Mutant^. This corresponds to the obvious enlargement of the internal cavity volume of CETP^Authentic^, which might provide a broad space for transfer. The significant structural changes in the “neck” may result from the residue replacement (C131A N88D N240D N341D) required for crystallization due to their close proximity. In MD simulations, residue Phe265 in CETP^Authentic^ orients towards residue Met433 at 50 ns, with d_Phe265-Met433_ and angle being 17.4 Å and 77.7°. Note that the angle is defined with the extreme edge carbon of a side-chain of residue Phe265 and the C_α_ atoms of residues Phe265 and Met260. After ~70 ns, d_Phe265-Met433_ slightly decreases to reach a plateau (17.0 Å), with an angle of 71.6°. At 80 ns, residue Phe265 orients towards the C-terminal end, and residues Phe265 and Met433 are pulled apart, giving rise to an open wider duct. During that time, d_Phe265-Met433_ and angle changed to 17.6 Å and 45.7°. These structural alterations in CETP^Authentic^ might be beneficial for lipid transfer. In CETP^Mutant^, d_Phe265-Met433_ has a smaller contraction, but the orientation of residue Phe265 remains unchanged over the 100 ns MD simulations. These results demonstrate that CETP^Authentic^, rather than CETP^Mutant^, facilitates the required conformational changes (continuous tunnel), corroborating the tunnel mechanism.

### 2.3. Virtual Screening for CETP

Previous CETP inhibitors were found to bind near the narrowing neck of the hydrophobic central tunnel, potentially hindering the connection between the N- and C-terminal pockets [30]. However, clinical study has been terminated due to poor efficacy against ASCVD. Previous studies discovered that CETP forms a transfer channel mediating CE from donor to acceptor lipoproteins by interaction of the N-terminus with HDL and the C-terminus with LDL or VLDL. Taking into account the flexibility of the N- and C-terminal domains, virtual screening approaches were employed targeting the N- and C-terminal β-barrel domains of CETP^Authentic^ to discover the potential CETP inhibitors from the bioactive compounds in the ZINC database via the cDocker algorithm [40]. Based on cDocker interaction energies (*E*_int_) of both N- and C-termini, the top eight ligands were selected for further analysis, including ZINC000002010603, ZINC000006248133, ZINC000005871812, ZINC000002261174, ZINC000003526223, ZINC000005871644, ZINC000007067674, and ZINC000006242926. Subsequently, each docked complex was further refined by 100 ns MD simulations. 

As shown in Appendix A, each docked complex reaches fundamental convergence after ~50 ns, indicated by the time evolutions of the backbone-atom RMSDs. The representative conformation of eight docked complexes, extracted from MD trajectories of ligands with CETP N-terminal and C-terminal domains, respectively, indicates that the binding positions of the ligands have changed slightly through the MD simulations. The complexes formed between ligands and the CETP N-terminal domain exhibit H-bonding interactions between the oxygen atoms of ligand ZINC000002010603 and the sulfur atoms of ligands ZINC000005871812 and ZINC000007067674 with residue Gln111. In addition, the ligand ZINC000006248133 has an H-bonding interaction with residue Trp106. These hydrogen bonding interactions play a role in stabilizing the Ω5 region of the N-terminal domain. The ligands mainly have hydrophobic interactions with certain hydrophobic residues Leu52 in the Ω4 region, Ala104-Ile109 in the Ω5 region, and Trp162 in the α-helix structure of the N-terminal domain (Appendix A). For the complexes of ligands and the CETP C-terminal domain, ZINC000005871812 has an H-bonding interaction with Leu285. ZINC000003526223 and ZINC000007067674 have H-bonding interactions with Arg424. All the ligands mainly have hydrophobic interactions with certain residues Leu285–Leu296 near the Ω1 region and Met412–Arg424 in the α-helix structure. Additionally, ZINC000006248133, ZINC000003526223, ZINC000005871644, ZINC000007067674, and ZINC000006242926 also have hydrophobic interactions with residues in the C-terminal β-barrel region (Appendix A). Furthermore, the binding free energies (Δ*G_bind_*) of CETP N-terminus and ZINC000002010603, ZINC000006248133, ZINC000005871812, ZINC000002261174, ZINC000003526223, ZINC000005871644, ZINC000007067674, and ZINC000006242926, which encompass the electrostatic energy (Δ*E*_ele_), van der Waals interactions (Δ*E*_vdw_), polar solvation energy (Δ*G*_GB_), and non-polar solvation energy (Δ*G*_sur_), are summed to −23.6 ± 0.2, −25.4 ± 2.1, −25.4 ± 2.1, −35.0 ± 1.7, −0.2 ± 0.2, −26.3 ± 2.3, −19.7 ± 2.8, and −32.6 ± 1.5 kcal·mol^−1^, respectively (Table 2). The binding free energies (Δ*G_bind_*) of CETP C-terminus and ZINC000002010603, ZINC000006248133, ZINC000005871812, ZINC000002261174, ZINC000003526223, ZINC000005871644, ZINC000007067674, and ZINC000006242926 are summed to −32.9 ± 2.3, −38.8 ± 1.6, −45.3 ± 1.3, −33.5 ± 1.8, −46.5 ± 1.3, −39.4 ± 1.4, −43.4 ± 1.0 and −42.2 ± 1.4 kcal·mol^−1^, respectively (Table 3). Based on the binding energy calculated by the MM/GBSA method, the ligand ZINC000006242926, which demonstrates strong binding affinity to both the N-terminal and C-terminal domains, was selected for further structural dynamics analysis.

### 2.4. Structural Dynamics Analysis upon Ligand Binding

Upon binding of ligand ZINC000006242926 to the CETP N-terminal domain, the value of d_Trp106-Trp162_ decreases significantly from the initial 16.04 Å to 13.22 Å and the value of d_Phe301-Met412_ exhibits minor variations, consistently hovering around 8 Å, while the d_Phe265-Met433_ remains nearly unchanged (Figure 5). The d_Phe265-Met433_ distance undergoes minimal change, decreasing from 17.87 Å to 17.50 Å. This observation suggests that the small molecule, being distally positioned, exerts a relatively weaker influence on the neck region. The analysis conducted using the Fpocket program [41] reveals a transformation in the central cavity of CETP, shifting from a continuous internal cavity volume of approximately 4898 Å^3^ to a larger main cavity with a volume of 4104 Å^3^, accompanied by the presence of smaller cavities. When ligand ZINC000006242926 binds to the C-terminus, the effects on the N-opening and C-opening are even greater, with d_Trp106-Trp162_ decreasing to 12.7 Å and d_Phe301-Met412_ decreasing to 7.94 Å, which is much smaller than the 10 Å distance required for CE to enter LDL or VLDL smoothly [5] (Figure 6). In addition, residue Phe265 exhibits a slight tilt towards the C-terminus and the d_Phe265-Met433_ distance undergoes minimal change, decreasing from 17.75 Å to 16.39 Å. The central cavity of CETP separates into two distinct broken cavities with volumes of 2284 Å^3^ and 2013 Å^3^, respectively, from a continuous cavity. The sizes of both the N-opening and C-opening are significantly reduced, suggesting that the ligand may inhibit the binding of both N-terminal and C-terminal domains to HDL and LDL or VLDL. The binding of small ligands to CETP may form a spatial site block and interfere with the formation of continuous tunnels, thus inhibiting CE transport from HDL to CETP or diffusion from CETP to the water phase. 

PCA analyses of systems CETP^Authentic^, CETP N-terminus–ZINC000006242926, and CETP C-terminus–ZINC000006242926 were performed to characterize the dynamic conformational changes in CETP structure induced by the ligand. The dynamic changes in CETP are relatively slight during the MD simulation of the CETP^Authentic^ system, in which the N-terminal domain has a slight tendency to move away from the concave surface, while the C-terminal domain has a tendency to move closer to the concave surface (Figure 7A), consistent with the conformational analysis shown in Appendix A. Upon binding of ligand ZINC000006242926 to the N-terminal domain, the N-terminal domain and part of the C-terminal domain have a tendency to move toward the concave surface, which is relatively small (Figure 7B). Notably, the movement of Helix X tending toward the N-terminus is obvious. Electron microscopy studies of the binary complex CETP–HDL showed that the CETP N-terminal domain can penetrate about 50 Å into HDL and can penetrate the HDL surface and core. Helix X, located approximately 50~60 Å from the end of the N-terminal structural domain, may also be involved in lipoprotein sensing and attachment [5]. And experimental studies have shown that mutant CETP lacking Asp470-Leu475, although able to penetrate HDL to some extent, is defective in achieving maximal lipid transfer function, verifying that Helix X may perform a docking function with HDL and allow more durable CE transfer [28]. By contrast, the alteration in movement tendency and flexibility of Helix X induced by the binding of ZINC000006242926 may affect CETP function in transferring CE. When the ligand ZINC000006242926 binds to the C-terminal domain, both the N- and C-terminal domains of CETP exhibit an obvious tendency to move toward the concave surface, and the “neck” region shows a tendency to move toward the convex surface, resulting in the curvature of the CETP convex surface increasing significantly (Figure 7C). Atomistic MD simulations have confirmed that the isolated central cavities connect and form a continuous tunnel by rotation in the β-barrel direction for lipid transport [20,42]. The increased curvature of CETP may disrupt the tunnel cavity structure and hinder CE transport. On the other hand, the shuttle model suggests that CETP, the intrinsic curvature of which matches well with the curvature of HDL particles, attaches to the surface of lipoproteins through the concave surface. The two tunnel openings on the concave surface are thought to be channels for the flow of neutral lipids between the particles [9,11], while the increased curvature of CETP may limit its binding to HDL and lipid exchange. 

The structural dynamics of CETP induced by ligand ZINC000006242926 are further analyzed using the dynamic cross-correlation map or matrix (DCCM). In the DCCM plot, blue indicates a significant positive correlation, red indicates obvious anti-correlation, and white regions indicate less correlation. The binding of ZINC000006242926 to the CETP N-terminus significantly weakens the negative correlation between residues Lys180–Pro246 located near the “neck” region and the Ω4, Ω5, and Ω6 regions (Appendix A), probably due to the hydrophobic interactions between ZINC000006242926 and residues Leu52, Leu107, and Trp162 (Appendix A) stabilizing this region. This phenomenon also occurs in the MD simulations of the ZINC000006242926–CETP C-terminus system (Appendix A). In addition, the positive correlation of the Ω1 region with Ω4 and Ω5 almost disappears when the ligand is bound to the CETP C-terminus, probably owing to the hydrophobic interactions of the ligand and the residues Glu291, Ala294, and Val295 near the Ω1 region stabilizing the region (Appendix A). Residues Phe429–Ser476 including the Helix X structure show an enhanced negative correlation with residues Ser343–Asp366 in β-barrel around the Ω2 region and Thr393-Pro419 in the α-helical structure (Appendix A), indicating that the Helix X structure moves in the opposite direction to the C-terminal domain of CETP, and the results are consistent with PCA analysis (Figure 7C). The correlations of Helix X with other residues of CETP, including positive and negative correlations, are significantly weaker in both complexes.

Previous research has shown that the hydrophobic interactions between the CETP inhibitors torcetrapib and anacetrapib and the CETP tunnel center may alter the flexibility of CETP, the structure of the distal β-barrel domain, and the structure–function relationship between phospholipids and Helix X, resulting in the inhibitors blocking the connection between the N-terminal and C-terminal pockets and disrupting the interaction of CETP with lipoproteins to reduce the transport of CE [20,22,23,30,34]. The screened ligand ZINC000006242926 can bind to both the N-terminal and C-terminal domains, altering the curvature of CETP and the dynamic cross-correlation between important residues, in addition to having an effect on CETP flexibility as well as the structure of both ends and helix X. According to the analysis, ZINC000006242926 may be a potential CETP inhibitor that warrants further experimental validation.

## 3. Materials and Methods

### 3.1. System Preparation

The boomerang model of CETP with 2.2 Å resolution (accession number 2OBD) was used as a starting structure [11]. For convenience, the structure of the mutant c444 construct (C1A C131A N88D N240D N341D) is referred to as “CETP^Mutant^” throughout this work. To obtain the authentic structure (hereafter called “CETP^Authentic^”, Figure 1), the missed four residues (C1 S2 K3 G4) were recruited, and the mutated residues were reversed using Discovery Studio 2019 Client (Waltham, MA, USA) [43,44]. In accordance with previous studies [13,45], all the hetero-atoms were removed using Discovery Studio 2019 Client [43,44], and missing hydrogen atoms were added based on the expected charge distributions of amino acids at pH 7.4 using the H++ server [13,43,46]. The two initial models were then optimized with the conjugated gradient (CG) method until they converged to 0.01 kcal mol^−1^ Å^−1^ [17]. 

Correct ionization and low-energy conformers of the ‘Drugs-Now’ subset of the ZINC database [32] were obtained by Discovery Studio and the CHARMm force field [47]. The virtual screening process was performed via the cDocker algorithm [40] and the representative CETP equilibrium conformation was derived from our MD simulations [31,48]. The details of receptor-based screening agree with our previous works [49,50]. Briefly, the binding sites of receptors were assigned with a sphere of 10.0 Å at the N-terminal and C-terminal domains, respectively, and the optimal orientations of compounds within proteins were probed on the basis of interactions with binding residues and geometrical matching qualities. The optimal docked complexes were further selected to be energy-minimized using the conjugate gradient (CG) method until converged to 0.01 kcal mol^−1^ Å^−1^.

### 3.2. MD Simulations

Each system was sufficiently equilibrated by 100.0 ns MD simulations, using AMBER18 software (San Francisco, CA, USA) [51,52] and AMBER ff14SB [53], as previously recommended [54]. Details of the MD simulation setup agree with references [55,56]. Briefly, each system was solvated in a cubic box of water molecules extending at least 10.0 Å from any solute atom and the TIP3P water model was used in the simulation. Na^+^ counter-anions were placed to neutralize the system [13]. The steepest descent (SD) and conjugate gradient (CG) methods were utilized to eliminate poor contacts in the initial structures. Then, a gradual heating process was applied to each system, raising the temperature from 0 to 310 K within 1.0 ns. Subsequently, the systems underwent further equilibration in a canonical ensemble (NVT) at 310 K for 1.0 ns with 2 kcal mol^−1^ Å^−2^ position restraints on the backbone. Finally, the systems were equilibrated in an isothermal–isobaric ensemble (NPT) at 310 K and 1 Bar. All MD simulations were performed using periodic boundary conditions, and the cutoff radius for coulomb and van der Waals interactions was set to 8.0 Å. The simulation time step of 2.0 fs was used, and coordinates were collected every 10.0 ps. In order to improve sampling and collect more data, each system was repeated three times, with different random number generator seeds each time.

### 3.3. Analysis

The cpptraj module of Amber Tools 18 was used to perform the dynamic analysis [52]. Fpocket program [41] was employed to characterize the internal cavity volume of CETP. Principal component analysis (PCA) performed on the Cα atoms and dynamic cross-correlation matrices (DCCM) [57] were used to understand the conformational changes upon ligand binding and calculate average correlations between the motion of atoms in protein [58,59]. Discovery Studio 2019 Client was used for structural plotting and visualization [43]. 

All values of binding free energies (Δ*G_bind_*) were calculated using the molecular mechanics–generalized Born surface area method (MM/GBSA) which can be used for various drug–ligand systems without additional regression [60,61]. The binding free energy is estimated from the energies of the protein, ligand, and complex.
Δ*G_bind_* = Δ*G_complex_* − Δ*G_protein_* − Δ*G_ligand_*(1)

The binding free energy (Δ*G_bind_*) consists of electrostatic energy (Δ*E*_ele_), van der Waals (Δ*E*_vdw_), polar solvation energy (Δ*G*_GB_), non-polar solvation energy (Δ*G*_sur_), and entropy contribution (−*T*Δ*S*) [62]. The values were evaluated with 500 snapshots evenly extracted from 50~100 ns MD trajectories. 

## 4. Conclusions

Numerous X-ray, EM, and MD studies conducted thus far have revealed some important physical attributes of CETP and the lipid-exchange mechanisms behind these behavioral effects. However, their conclusions are based on artificial CETP (CETP^Mutant^, with four mutated residues and four missed residues), constructed for protein crystallization. In this work, comprehensive MD simulations of CETP^Authentic^ and CETP^Mutant^ were performed to evaluate their structural differences influencing CETP-mediated lipid exchange, including stability, flexibility, hydrophobicity, and residue reorientation.

Our explicit solvent MD simulations show that CETP^Authentic^ is more flexible than CETP^Mutant^, with the overall good agreement of experimental results that the native CETP is difficult to crystallize. CETP^Authentic^ is much more flexible in the regions of flap Ω3 and Helix X, and its C-terminal domain partly flexes back to Helix X; CETP^Mutant^, meanwhile, is more stable, and its N-terminal domain stretches longer along the axis. Their internal hydrophobic cavities are both enlarged during the MD simulations, especially the authentic one. Further structural analyses reveal that the configuration change of the N-terminal end (flaps Ω4–6) is similar in CETP^Authentic^ and CETP^Mutant^, supporting the existence of an N-opening. The configurations of the C-terminal distal region (flaps Ω1–3) in CETP^Authentic^ and CETP^Mutant^ are different in aqueous solutions, with considerable differences in hydrophobic SASA. The authentic one retains even more hydrophobicity and is more likely to form an opening, with a larger maximum distance between residues Phe301 and Met412 (d_Phe301-Met412_), which might be the key residues for the formation of a “C-opening”. In all MD systems, Helix X in either CETP^Authentic^ or CETP^Mutant^ adopts the closed state, and this motion should be a necessary condition for lipid exchange. Regarding the “neck” region, CETP^Authentic^ rather than CETP^Mutant^ facilitates the formation of a continuous tunnel, with a constant distance between residues Phe265 and Met433 (d_Phe265-Met433_). In CETP^Authentic^, residue Phe265 orients away from residue Met433 and provides a more conducive environment for lipid exchange to overcome the potential barrier formed by residues Phe265 and Met433. In short, there are obvious spatial distinctions between authentic CETP and the mutant construct. CETP^Authentic^, rather than CETP^Mutant^, supports the tunnel mechanism, and further studies on the mechanisms of CETP-mediated lipid exchange should fully consider this point of view.

Cryo-EM and MD results indicated that the N-terminal and C-terminal structures are flexible and have important roles in the lipid-transfer function. Eight small molecules were selected as potential ligands, which bind to the N- and C-terminal domains of CETP. Based on the complicated MD simulations and MM/GBSA calculations, the ligand ZINC000006242926 was chosen for further investigation to assess its influence on the structural dynamics of CETP. The mutant structures of CETP in complex with the inhibitor torcetrapib, which has shown a significant effect on plasma lipoprotein levels in clinical trials, as well as being an analog of structurally different inhibitor families, have revealed that the inhibitors bind to the protein by forming hydrophobic interactions, thereby obstructing the connection between the N- and C-terminal pockets [11]. Furthermore, we have found that the binding of ligands reduces the size of the N-opening and C-opening and disrupts the formation of the continuous tunnel, which may limit the transport function of CETP to CE. In addition, the motion tendency of Helix X is also altered and is more pronounced when ligands bind to the N-terminus, suggesting that it may affect the function of CETP. The ligand could increase the curvature of CETP concave surface bending, especially when it binds at the C-terminus, which is detrimental to the CETP transport of lipids. All these results provide a theoretical basis and application guidance for subsequent experimental validation and development of CETP inhibitors. 

## Figures and Tables

**Figure 1 ijms-24-12252-f001:**
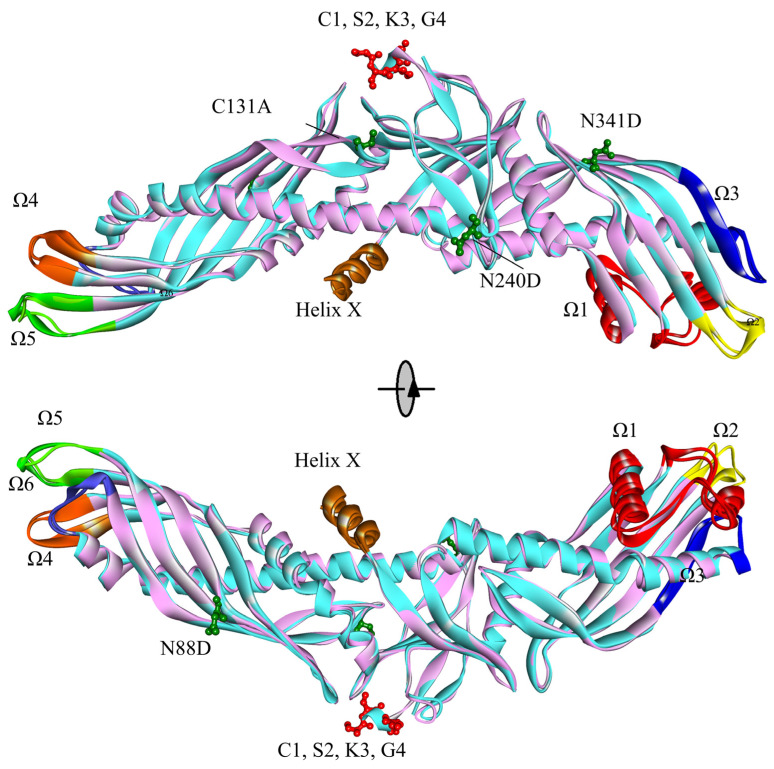
Overall initiating structures of CETP^Mutant^ (cyan) and CETP^Authentic^ (pink). Four mutated residues including C131A N88D N240D N341D (green) and four missed residues including C1, S2, K3, and G4 (red) are presented by ball and stick models, respectively; the regions of Ω1–6 and Helix X are shown in distinct colors: Ω1 (red), Ω2 (yellow), Ω3 (blue), Ω4 (green), Ω5 (indigo), Ω6 (orange), and Helix X (brown).

**Figure 2 ijms-24-12252-f002:**
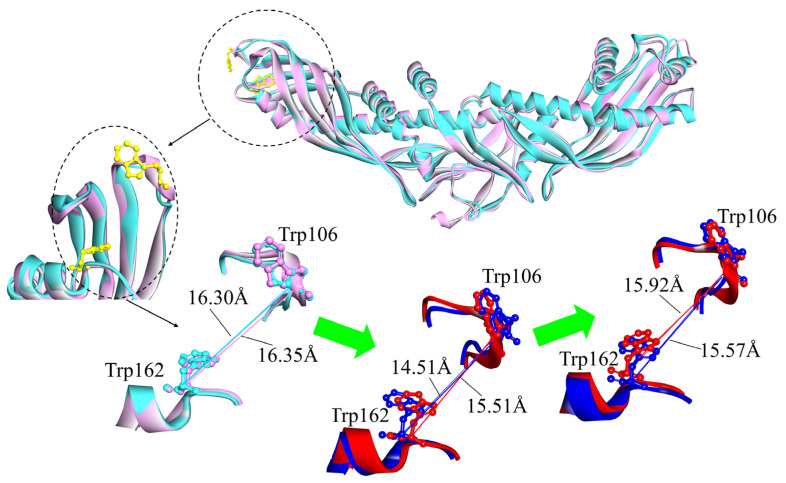
Superimposition of the N-opening regions in CETP^Mutant^ and CETP^Authentic^. Initiating conformations (0 ns) are shown in cyan (mutant) and pink (authentic), while representative conformations (30 and 80 ns) are in blue (mutant) and red (authentic). Residues Trp106 and Trp162 are presented by ball and stick models. In the MD simulations, maximum distances between Trp106 and Trp162 are changed from 16.30 Å and 16.35 Å to 14.51 and 15.51 Å, and then reached the final values of 15.57 and 15.92 Å.

**Figure 3 ijms-24-12252-f003:**
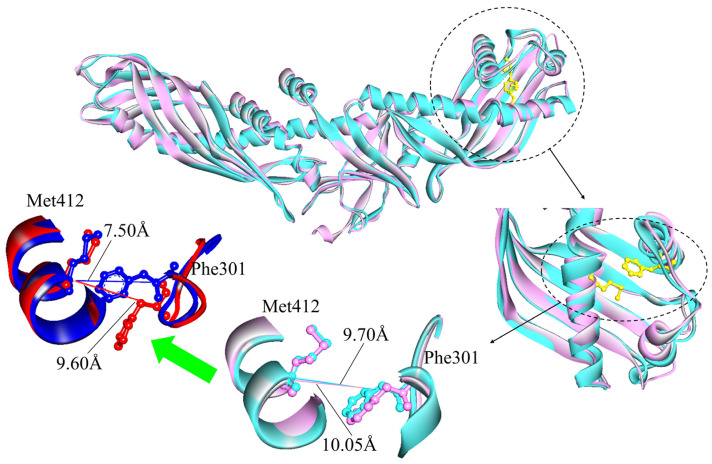
Superimposition of the “C-opening” regions in CETP^Mutant^ and CETP^Authentic^. Initiating conformations (0 ns) are shown in cyan (mutant) and pink (authentic), while representative conformations (25 ns) are in blue (mutant) and red (authentic). Residues Phe301 and Met412 are presented by ball and stick models. In the MD simulations, maximum distances between Phe301 and Met412 are changed from 9.70 and 10.05 Å to 7.50 and 9.60 Å, respectively.

**Figure 4 ijms-24-12252-f004:**
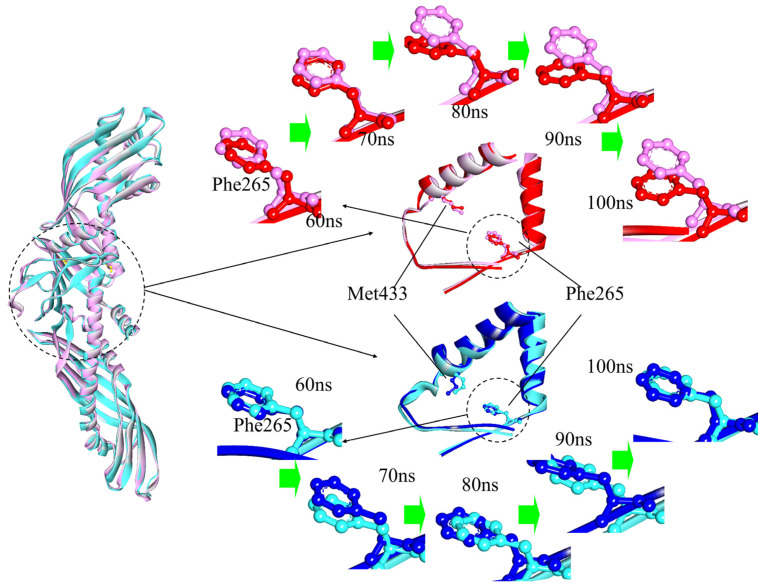
Superimposition of the “neck” regions in CETP^Mutant^ and CETP^Authentic^. Initiating conformations, which are selected as comparisons (50 ns), are shown in cyan (mutant) and pink (authentic), while the representative conformations (60, 70, 80, 90, and 100 ns) are in blue (mutant) and red (authentic). Residues Phe265 and Met433 are presented by ball and stick models.

**Figure 5 ijms-24-12252-f005:**
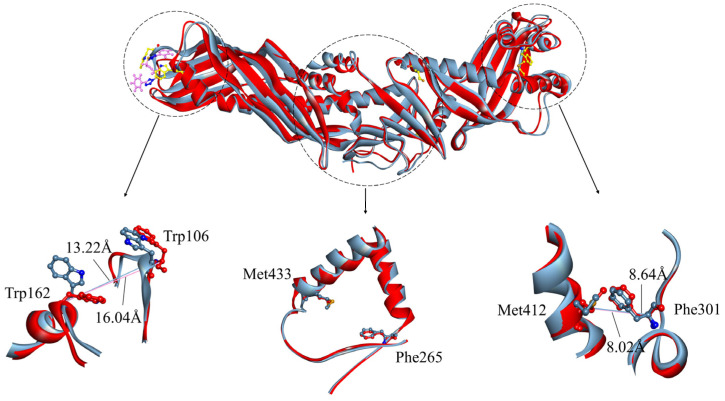
Comparison of CETP structure after binding of ligand ZINC000006242926 to the N-terminal domain of CETP. The initial representative conformation is shown in red and the final representative conformation is shown in blue. Superimposition of the “N-opening“, “neck“, and “C-opening” regions is shown, wherein residues Trp106 and Trp162, Phe301 and Met412, and Phe265 and Met433 are presented by ball and stick models.

**Figure 6 ijms-24-12252-f006:**
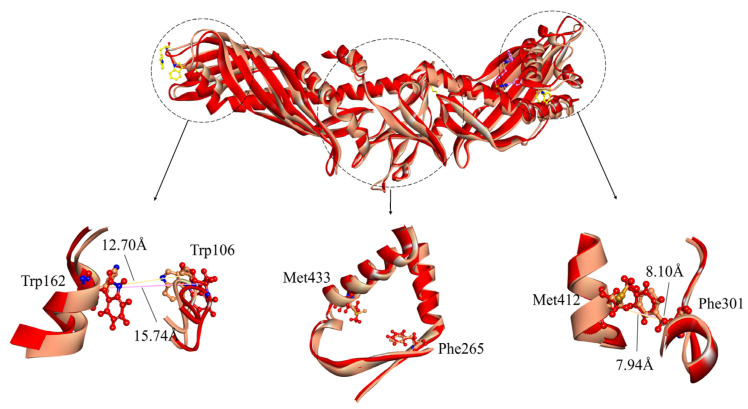
Comparison of CETP structure after binding of ligand ZINC000006242926 to the C-terminal domain of CETP. The initial representative conformation is shown in red and the final representative conformation is shown in tan. Superimposition of the “N-opening“, “neck“, and “C-opening” regions is shown, wherein residues Trp106 and Trp162, Phe301 and Met412, and Phe265 and Met433 are presented by ball and stick models.

**Figure 7 ijms-24-12252-f007:**
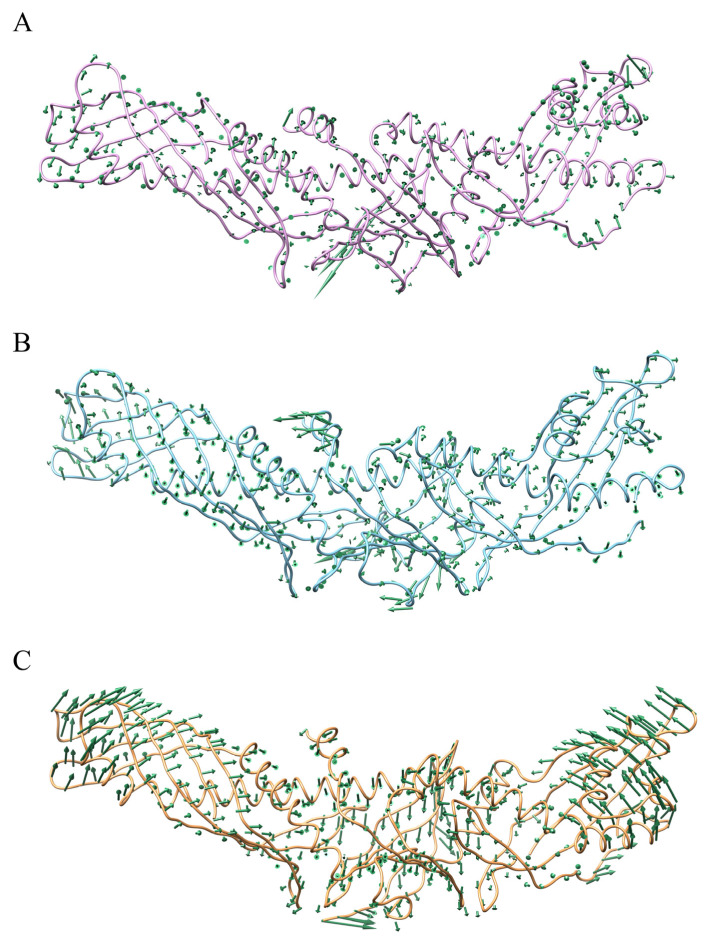
Vector field representations of the first principal component (PC) obtained for CETP in (**A**) CETP^Authentic^, (**B**) CETP N-terminus–ZINC000006242926, and (**C**) CETP C-terminus–ZINC000006242926.

**Table 1 ijms-24-12252-t001:** Distribution alterations (%) in hydrophilic/hydrophobic solvent-accessible surface areas (SASA) between the initiating and equilibrium conformations ^1^.

	CETP^Mutant^			CETP^Authentic^		
Region	Hydrophobic	Hydrophilic	Total	Hydrophobic	Hydrophilic	Total
N-end ^2^	−0.05	0.02	−0.04	−0.05	0.00	−0.02
C-end ^3^	−0.02	0.08	0.05	0.07	0.08	0.08
Ω1	−0.13	0.09	−0.05	0.04	0.04	0.04
Ω2	0.15	0.03	0.18	0.19	0.18	0.19
Ω3	0.05	0.08	0.12	0.05	0.10	0.09
Ω4	0.08	0.05	0.13	0.04	0.01	0.02
Ω5	−0.08	−0.07	−0.15	−0.08	−0.06	−0.07
Ω6	−0.15	0.07	−0.08	−0.10	0.05	0.00
Helix X	0.02	0.03	0.06	0.10	0.01	0.03

^1^ The values of initiating conformations are used as the benchmarks; ^2^ N-terminal end consists of Ω 4~6; ^3^ C-terminal end consists of Ω 1~3.

**Table 2 ijms-24-12252-t002:** Binding free energies (Δ*G_bind_*) and their components of CETP^Authentic^ N-terminus and ligands ^1^.

ZINC ID	Δ*E*_ele_ ^2^	Δ*E*_vdw_ ^3^	Δ*G*_sur_ ^4^	Δ*G*_GB_ ^5^	Δ*G_bind_*
ZINC000002010603	−2.2 ± 1.5	−14.2 ± 0.9	−8.9 ± 0.5	1.7 ± 0.1	−23.6 ± 0.2
ZINC000006248133	−5.5 ± 2.0	−12.6 ± 1.0	−8.4 ± 0.7	1.1 ± 1.5	−25.4 ± 2.1
ZINC000005871812	−5.6 ± 1.8	−13.1 ± 1.1	−8.7 ± 0.7	2.0 ± 1.3	−25.4 ± 2.1
ZINC000002261174	−5.8 ± 1.4	−19.5 ± 0.9	−12.3 ± 0.6	2.5 ± 1.0	−35.0 ± 1.7
ZINC000003526223	0.0 ± 0.5	−0.1 ± 0.9	−0.1 ± 0.7	−0.1 ± 0.4	−0.2 ± 0.2
ZINC000005871644	−2.6 ± 1.4	−14.9 ± 1.2	−9.7 ± 0.8	0.9 ± 1.1	−26.3 ± 2.3
ZINC000007067674	−2.2 ± 1.4	−11.3 ± 1.5	−7.0 ± 0.1	0.7 ± 1.1	−19.7 ± 2.8
ZINC000006242926	−2.5 ± 1.1	−19.5 ± 0.8	−11.2 ± 0.5	0.6 ± 0.9	−32.6 ± 1.5

^1^ All values, along with their standard deviations (S.D.), are expressed in kcal mol^−1^. ^2^ Electrostatic energy. ^3^ Van der Waals interactions. ^4^ Non-polar solvation energy. ^5^ Polar solvation energy.

**Table 3 ijms-24-12252-t003:** Binding free energies (Δ*G_bind_*) and their components of CETP^Authentic^ C-terminus and ligands ^1^.

ZINC ID	Δ*E*_ele_ ^2^	Δ*E*_vdw_ ^3^	Δ*G*_sur_ ^4^	Δ*G*_GB_ ^5^	Δ*G_bind_*
ZINC000002010603	−5.8 ± 3.1	−17.5 ± 1.1	−10.1 ± 0.7	0.4 ± 2.7	−32.9 ± 2.3
ZINC000006248133	−9.1 ± 2.3	−19.6 ± 0.8	−13.0 ± 0.6	3.2 ± 2.0	−38.8 ± 1.6
ZINC000005871812	−2.8 ± 1.6	−26.6 ± 0.8	−15.5 ± 0.4	−0.4 ± 1.3	−45.3 ± 1.3
ZINC000002261174	−2.7 ± 1.3	−19.0 ± 0.9	−11.5 ± 0.6	−0.3 ± 1.1	−33.5 ± 1.8
ZINC000003526223	−7.0 ± 1.7	−28.5 ± 0.8	−16.5 ± 0.5	5.5 ± 1.6	−46.5 ± 1.3
ZINC000005871644	−5.1 ± 1.6	−23.1 ± 0.8	−14.3 ± 0.5	3.1 ± 1.4	−39.4 ± 1.4
ZINC000007067674	−2.3 ± 1.2	−26.1 ± 0.6	−15.6 ± 0.3	0.6 ± 1.1	−43.4 ± 1.0
ZINC000006242926	−1.4 ± 1.2	−25.9 ± 0.9	−15.0 ± 0.5	0.1 ± 1.1	−42.2 ± 1.4

^1^ All values, along with their standard deviations (S.D.), are expressed in kcal mol^−1^. ^2^ Electrostatic energy. ^3^ Van der Waals interactions. ^4^ Non-polar solvation energy. ^5^ Polar solvation energy.

## Data Availability

Computational instructions and the data of this work are provided in the main text and supporting information; further information and requests may be directed to and will be fulfilled by Zhiwei Yang (yzws-123@xjtu.edu.cn), the lead contact. Software used: BIOVIA Discovery Studio 2019 Client, https://www.3ds.com/, accessed on 8 March 2023; Amber18, http://ambermd.org/, accessed on 8 March 2023; AmberTools18, http://ambermd.org/AmberTools.php, accessed on 8 March 2023.

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
