# Peer review of "Dissecting the Structural Dynamics of Authentic Cholesteryl Ester Transfer Protein for the Discovery of Potential Lead Compounds: A Theoretical Study"

_ijms, 2023, doi:10.3390/ijms241512252_

Round 1
Reviewer 1 Report
This manuscript "IJMS-2504151," presents a well-executed and exclusively computational study on the cholesteryl ester transfer protein (CETP). The authors conducted molecular dynamics simulations (MDs) on both the crystallized mutant (CEPTCrystal) and the native (CEPTNascent) counterparts to elucidate the structural differences. Additionally, the authors performed a screening of the ZINC database to identify potential ligands that could serve as inhibitors of the enzyme.
The computational methods employed in this study are adequately described and effectively implemented using suitable software such as AMBER and Discovery Studios. The parameterization of the systems under investigation is accurate, and the overall work is valid. The results and discussion sections offer satisfactory insights.
There are a few minor revisions that could be made:
· In the abstract, on line 11, rephrase the sentence "Current simulation and experiment ..." to enhance clarity.
· Besides, the word “experiment” on line 11 should be omitted as this is a pure computational work.
· In the abstract, on line 17, rephrase the sentence "Considering the structural disparities characteristics ..." to something like "Considering the characteristics of structural disparities ..."
· In theresults and discussion section, it is recommended to provide a more explicit explanation of the energy component terms mentioned in the footnote of Table 2. This explanation should align with the description provided in lines 480-481 of the Materials section.
· Also the footnote can be changed to “All values, along with their standard deviations (S.D.), are expressed in kcal mol-1.”
· In the methods section, on lines 437-439, clarify which program was used for the modifications involving all hetero-atoms at pH 7.4.
· On line 439 correct the typo error from "pH7.4" to "pH 7.4."
· On line 482 of the methods section, clarify whether the poses calculated in 500 snapshots correspond to the equilibration time of the system.
· Citation [15] is missing.
In order to provide a more comprehensive understanding of this study, it would be valuable to conduct experiments involving the 8 ligands and the CEPT protein to validate the computational findings. Such experimental confirmation could make the manuscript more suitable for publication in IJMS.
Minor editing of English language required
Author Response
This manuscript "IJMS-2504151," presents a well-executed and exclusively computational study on the cholesteryl ester transfer protein (CETP). The authors conducted molecular dynamics simulations (MDs) on both the crystallized mutant (CEPTCrystal) and the native (CEPTNascent) counterparts to elucidate the structural differences. Additionally, the authors performed a screening of the ZINC database to identify potential ligands that could serve as inhibitors of the enzyme.
The computational methods employed in this study are adequately described and effectively implemented using suitable software such as AMBER and Discovery Studios. The parameterization of the systems under investigation is accurate, and the overall work is valid. The results and discussion sections offer satisfactory insights.
Response: We sincerely appreciate the reviewer for his/her careful reading.
There are a few minor revisions that could be made:
Comment 1: In the abstract, on line 11, rephrase the sentence "Current simulation and experiment ..." to enhance clarity.
Besides, the word “experiment” on line 11 should be omitted as this is a pure computational work.
Response 1: Thanks for reviewer’s comments. The initial structure of CETP used in the MD simulations was taken from the crystal structure, and the structural and dynamic characteristics of CETP were analyzed via the MD simulations. With respect to the reviewer’s comment, it has been revised as “Current structural and functional investigations on cholesteryl ester transfer protein (CETP) inhibitor design are nearly entirely taken from a fully active mutation (CETPCrystal) constructed for protein crystallization.” (lines 11-13 of the abstract section)
Comment 2: In the abstract, on line 17, rephrase the sentence "Considering the structural disparities characteristics ..." to something like "Considering the characteristics of structural disparities ..."
Response 2: Thanks for reviewer’s comments. We have updated the contents in the revised manuscript in the abstract, on line 17-20. " Given the structural differences between the N- and C-terminal domains of the CETPNascent and CETPCrystal, as well as the importance of the two domains in the lipid transfer process, we performed a virtual screening for the N- and C-terminal domains, respectively, to identify potential lead compounds targeting CETP."
Comment 3: In the results and discussion section, it is recommended to provide a more explicit explanation of the energy component terms mentioned in the footnote of Table 2. This explanation should align with the description provided in lines 480-481 of the Methods section.
Also the footnote can be changed to "All values, along with their standard deviations (S.D.), are expressed in kcal mol-1."
Response 3: Thanks for reviewer’s comments. We have updated and added the explicit explanation of the energy component terms in the footnote of Table 2 and Table 3. “1 All values, along with their standard deviations (S.D.), are expressed in kcal mol-1. 2 Electrostatic energy. 3 Van der Waals interactions. 4 Non-polar solvation energy. 5 Polar solvation energy. “
We have updated the contents in the revised manuscript on line 321-327 to correspond with what is mentioned in the Methods section. “Furthermore, the binding free energies (ΔGbind) between CETP N-terminus and ZINC000002010603, ZINC000006248133, ZINC000005871812, ZINC000002261174, ZINC000003526223, ZINC000005871644, ZINC000007067674, ZINC000006242926, which encompass the electrostatic energy (ΔEele), van der Waals interactions (ΔEvdw), polar solvation energy (ΔGGB), and non-polar solvation energy (ΔGsur), are summed to −23.60±0.15, −25.39±2.11, −25.37±2.13, −35.02±1.68, −0.21±0.21, −26.26±2.25, −19.73±2.75 and −32.61±1.46 kcal·mol-1 respectively (Table 2).” In addtion, we have updated the contents in the revised manuscript in lines 494-497 of the Methods section. " The binding free energy (ΔGbind) consists of electrostatic energy(ΔEele), van der Waals (ΔEvdw), polar solvation energy (ΔGGB), non-polar solvation energy (ΔGsur) and entropy contribution (−TΔS) 69."
Ref:
(69) Genheden, S.; Ryde, U., The MM/PBSA and MM/GBSA methods to estimate ligand-binding affinities. Expert Opin Drug Dis 2015, 10 (5), 449-461.
Comment 4: In the methods section, on lines 437-439, clarify which program was used for the modifications involving all hetero-atoms at pH 7.4.
Response 4: Thanks for reviewer’s comments. We have updated the contents to “all the hetero-atoms were removed using Discovery studio client 49, 50, and missing hydrogen atoms were added based on the expected charge distributions of amino acids at pH 7.4 using the H++ server 14, 49, 52.”
Ref:
(14) Lei, D.; Zhang, X.; Jiang, S.; Cai, Z.; Rames, M. J.; Zhang, L.; Ren, G.; Zhang, S., Structural features of cholesteryl ester transfer protein: a molecular dynamics simulation study. Proteins 2013, 81 (3), 415-25.
(49) Accelrys Discovery Studio 3.1. http://accelrys.com.
(50) Sastry, G. M.; Adzhigirey, M.; Day, T.; Annabhimoju, R.; Sherman, W., Protein and ligand preparation: parameters, protocols, and influence on virtual screening enrichments. J Comput Aided Mol Des 2013, 27 (3), 221-34.
(52) Anandakrishnan, R.; Aguilar, B.; Onufriev, A. V., H++3.0: automating pK prediction and the preparation of biomolecular structures for atomistic molecular modeling and simulations. Nucleic Acids Research 2012, 40 (W1), W537-W541.
Comment 5: On line 439 correct the typo error from "pH7.4" to "pH 7.4."
Response 5: Thanks for reviewer’s comments. It has been corrected in this version.
Comment 6: On line 482 of the methods section, clarify whether the poses calculated in 500 snapshots correspond to the equilibration time of the system.
Response 6: Thanks for reviewer’s comments. We have updated the contents on line 496. “The values were evaluated with 500 snapshots evenly extracted from 50 ~ 100 ns MD trajectories.”
Comment 7: Citation [15] is missing.
Response 7: Thanks for reviewer’s comments. It has been added in this version.
Comment 8: In order to provide a more comprehensive understanding of this study, it would be valuable to conduct experiments involving the 8 ligands and the CEPT protein to validate the computational findings. Such experimental confirmation could make the manuscript more suitable for publication in IJMS.
Response 8: Thanks for reviewer’s comments. The crystal structures of CETP in complex with the inhibitor torcetrapib with a significant effect on plasma lipoprotein levels in clinical trials and an analog of structurally different inhibitor families have showed that the inhibitor binds to the protein by forming hydrophobic interactions, thereby obstructing the connection between the N- and C-terminal pockets 45. The ligands we screened primarily hydrophobize with CETP and spatially occupy the N- and C-terminal binding pockets, blocking the connection between the N- and C-terminal pockets, inducing structural changes in CETP, and even influenting on the binding of CETP to HDL and LDL or VLDL. The current study is mainly to compare the structural differences between CETPNascent and CETPCrystal, and to screen based on the structural difference characteristics. It can provide an initial guide for the theoretical analysis of the binding site for subsequent experimental validation.
We have updated the contents in the revised manuscript on line 535-549. “The crystal structures of CETP in complex with the inhibitor torcetrapib, which has shown a significant effect on plasma lipoprotein levels in clinical trials, and an analog of structurally different inhibitor families have revealed that the inhibitor binds to the protein by forming hydrophobic interactions, thereby obstructing the connection between the N- and C-terminal pockets 46. Furthermore, we found that the binding of ligand reduces the size of N-opening and C-opening and disrupts the formation of the continuous tunnel, which may limit the transport function of CETP to CE. …… All these results provide a theoretical basis and application guidance for subsequent ex-perimental validation and development of CETP inhibitors.”
Ref:
(46) Qiu, X. Y.; Mistry, A.; Ammirati, M. J.; Chrunyk, B. A.; Clark, R. W.; Cong, Y.; Culp, J. S.; Danley, D. E.; Freeman, T. B.; Geoghegan, K. F.; Griffor, M. C.; Hawrylik, S. J.; Hayward, C. M.; Hensley, P.; Hoth, L. R.; Karam, G. A.; Lira, M. E.; Lloyd, D. B.; McGrath, K. M.; Stutzman-Engwall, K. J.; Subashi, A. K.; Subashi, T. A.; Thompson, J. F.; Wang, I. K.; Zhao, H. L.; Seddon, A. P., Crystal structure of cholesteryl ester transfer protein reveals a long tunnel and four bound lipid molecules. Nat Struct Mol Biol 2007, 14 (2), 106-113.

Reviewer 2 Report
The authors aimed to elucidate whether structural differences between nascent CETP and the mutant c444 CETP construct. They evaluated the spatial distinctions between CETPNascent and CETPCrystal. They investigated the stability, flexibility, hydrophobicity and residue reorientation. They also performed virtual screening and MD simulations based on the nascent CETP structure against the ZINC library. The reported results contribute tot he better understanding of the CETP-mediated lipid transfer mechanisms.
The paper is well-written. The figures and the supplemental material are impressive. The presented results are extremely useful and provides the basis of understanding of the dysfunctions of the maturally occuring genetic variants of the CETP gene.
Suggestions:
1. In the introduction please 2 or 3 sentences about the CETP gene, which encodes this protein. Mention that pathogenic variants of the CETP gene can result in the development of rare monogenic disease with autosomal dominant inheritance. Please check https://www.omim.org/entry/118470?search=CETP&highlight=cetp
2. Please also describe in 1 or 2 sentences, that how many naturally occuring diseases causing variants of the CETP is known so far. For this please also check: https://www.ncbi.nlm.nih.gov/clinvar/?term=CETP%5Bgene%5D&redir=gene
3. Please in the discussion, if it relevant, please add a small paragraph about the association of the naturally ocurring pathogenic or likely pathogenic variants and the significant structural residues you identified in this paper.
Very nice paper. Congratulations
Author Response
The authors aimed to elucidate whether structural differences between nascent CETP and the mutant c444 CETP construct. They evaluated the spatial distinctions between CETPNascent and CETPCrystal. They investigated the stability, flexibility, hydrophobicity and residue reorientation. They also performed virtual screening and MD simulations based on the nascent CETP structure against the ZINC library. The reported results contribute tot he better understanding of the CETP-mediated lipid transfer mechanisms.
The paper is well-written. The figures and the supplemental material are impressive. The presented results are extremely useful and provides the basis of understanding of the dysfunctions of the maturally occuring genetic variants of the CETP gene.
Response: We sincerely appreciate the reviewer for his/her careful reading.
Comment 1: In the introduction please 2 or 3 sentences about the CETP gene, which encodes this protein. Mention that pathogenic variants of the CETP gene can result in the development of rare monogenic disease with autosomal dominant inheritance. Please check https://www.omim.org/entry/118470?search=CETP&highlight=cetp
Response 1: Thanks for reviewer’s comments. We have updated the contents in the revised manuscript in the introduction, on line 30-35. “Cholesteryl ester transfer protein (CETP) encoded by the CETP gene, a 476-residue-long plasma glycoprotein, facilitates the hetero exchanges of cholesteryl esters (CE) and triglycerides (TG) 1. The development of hyperalphalipoproteinemia 1, a mono-genic condition with autosomal dominant inheritance, has been linked to pathogenic variants in the CETP gene, highlighting the crucial function of CETP in high-density lipo-protein metabolism 2-4.”
Ref:
(1) Rader, D. J.; Tall, A. R., The not-so-simple HDL story: Is it time to revise the HDL cholesterol hypothesis? Nat Med 2012, 18 (9), 1344-1346.
(2) Takahashi, K.; Jiang, X. C.; Sakai, N.; Yamashita, S.; Hirano, K.; Bujo, H.; Yamazaki, H.; Kusunoki, J.; Miura, T.; Kussie, P.; Matsuzawa, Y.; Saito, Y.; Tall, A., A MISSENSE MUTATION IN THE CHOLESTERYL ESTER TRANSFER PROTEIN GENE WITH POSSIBLE DOMINANT EFFECTS ON PLASMA HIGH-DENSITY-LIPOPROTEINS. Journal of Clinical Investigation 1993, 92 (4), 2060-2064.
(3) Akita, H.; Chiba, H.; Tsuchihashi, K.; Tsuji, M.; Kumagai, M.; Matsuno, K.; Kobayashi, K., Cholesteryl ester transfer protein gene: two common mutations and their effect on plasma high-density lipoprotein cholesterol content. J Clin Endocrinol Metab 1994, 79 (6), 1615-8.
(4) Amberger, J. S.; Bocchini, C. A.; Schiettecatte, F.; Scott, A. F.; Hamosh, A., OMIM.org: Online Mendelian Inheritance in Man (OMIM (R)), an online catalog of human genes and genetic disorders. Nucleic Acids Research 2015, 43 (D1), D789-D798.
Comment 2: Please also describe in 1 or 2 sentences, that how many naturally occuring diseases causing variants of the CETP is known so far. For this please also check: https://www.ncbi.nlm.nih.gov/clinvar/?term=CETP%5Bgene%5D&redir=gene
Response 2: Thanks for reviewer’s comments. We have added the contents in the revised manuscript in the abstract, on line 92-93. “Approximately 30 CETP variations have been recognized as clinically harmful among the numerous CETP variants (https://www.ncbi.nlm.nih.gov/clinvar/) 27”
Ref:
(27) Landrum, M. J.; Lee, J. M.; Benson, M.; Brown, G. R.; Chao, C.; Chitipiralla, S.; Gu, B. S.; Hart, J.; Hoffman, D.; Jang, W.; Karapetyan, K.; Katz, K.; Liu, C. L.; Maddipatla, Z.; Malheiro, A.; McDaniel, K.; Ovetsky, M.; Riley, G.; Zhou, G.; Holmes, J. B.; Kattman, B. L.; Maglott, D. R., ClinVar: improving access to variant interpretations and supporting evidence. Nucleic Acids Research 2018, 46 (D1), D1062-D1067.
Comment 3: Please in the discussion, if it relevant, please add a small paragraph about the association of the naturally ocurring pathogenic or likely pathogenic variants and the significant structural residues you identified in this paper.
Response 3: Thanks for reviewer’s comments. The effect of CETP with mutation Ile443Trp has been mentioned in the Results and Disussion section. And structure studies related to the effects of naturally occurring pathogenic variants on the structure of CETPs are limited, so we briefly elucidate in the Results and Discussiosn to clarify the association of the naturally ocurring pathogenic or likely pathogenic variants and the significant structural residues you identified in this paper. We have added the contents in the revised manuscript in the Results and Discussion, on line 235-239. “The interesting thing is that mutations of residues 54, 106 at the N-terminus, and residue 318 at the C-terminus are associated with hyperalphalipoproteinemia 1, where Trp106 is an important residue for CETP-HDL interaction, and therefore these mutations may alter the structure and hence the function of CETP (https://www.ncbi.nlm.nih.gov/clinvar/) 27, 40.”
Ref:
(27) Landrum, M. J.; Lee, J. M.; Benson, M.; Brown, G. R.; Chao, C.; Chitipiralla, S.; Gu, B. S.; Hart, J.; Hoffman, D.; Jang, W.; Karapetyan, K.; Katz, K.; Liu, C. L.; Maddipatla, Z.; Malheiro, A.; McDaniel, K.; Ovetsky, M.; Riley, G.; Zhou, G.; Holmes, J. B.; Kattman, B. L.; Maglott, D. R., ClinVar: improving access to variant interpretations and supporting evidence. Nucleic Acids Research 2018, 46 (D1), D1062-D1067.
(40) Cilpa-Karhu, G.; Jauhiainen, M.; Riekkola, M. L., Atomistic MD simulation reveals the mechanism by which CETP penetrates into HDL enabling lipid transfer from HDL to CETP. Journal of Lipid Research 2015, 56 (1), 98-108.

Reviewer 3 Report
I see a paper is written for medical or purely biological journal, without a careful description of the methods. I also see serious drawbacks in the methodology. The authors use words like "flexibility", "structural dynamics" when comparing the two structures, without showing how does it follow from the results of the simulation. Just showing pictures is not enough to support their claims. Besides, the following sentence:
"The binding free energy is estimated from energies of the protein, ligand, and complex."
is absolutely improper -- the free energy is a macroscopic quantity, connected to the microscopic energies through the partition function, and not as a difference between the energies as in Eq.1. The authors seem to be unaware of statistical mechanics, or, at least, do not take care in their claims.
I do not recommend the publication for the above reasons.
